# Diffused Task-Agnostic Milestone Planner

**Mineui Hong, Minjae Kang, and Songhwai Oh**
Department of Electrical and Computer Engineering and ASRI
Seoul National University
mineui.hong@rllab.snu.ac.kr, minjae.kang@rllab.snu.ac.kr, songhwai@snu.ac.kr

## Abstract

Addressing decision-making problems using sequence modeling to predict future trajectories shows promising results in recent years. In this paper, we take a step further to leverage the sequence predictive method in wider areas such as long-term planning, vision-based control, and multi-task decision-making. To this end, we propose a method to utilize a diffusion-based generative sequence model to plan a series of *milestones* in a latent space and to have an agent to follow the milestones to accomplish a given task. The proposed method can learn control-relevant, low-dimensional latent representations of milestones, which makes it possible to efficiently perform long-term planning and vision-based control. Furthermore, our approach exploits generation flexibility of the diffusion model, which makes it possible to plan diverse trajectories for multi-task decision-making. We demonstrate the proposed method across offline reinforcement learning (RL) benchmarks and an visual manipulation environment. The results show that our approach outperforms offline RL methods in solving long-horizon, sparse-reward tasks and multi-task problems, while also achieving the state-of-the-art performance on the most challenging vision-based manipulation benchmark.

## 1 Introduction

Developing a general-purpose agent that can handle a variety of decision-making problems is a long-standing challenge in the field of artificial intelligence. In recent years, there has been a growing interest in using offline reinforcement learning (RL) to tackle this problem [48, 35, 49]. One of the main advantages of offline RL is its ability to leverage the data comprised of multiple sub-tasks by stitching together the relevant transitions to perform a new and more complex task [8]. This property of offline RL allows an agent to learn diverse behaviors demonstrated by a human or other robots performing similar tasks. In order to stitch the task-relevant sub-trajectories from the undirected multi-task data, the majority of the offline RL methods rely on temporal difference learning which measures the value of the policy by bootstrapping the approximated value functions. However, despite the successes in the online setting which allows an agent to actively collect data [30, 37, 14], the bootstrapping method often causes instability in offline learning. To address this problem, the offline RL methods require special treatments such as using batch-constrained policies [11, 32, 25] and penalizing the value of out-of-distribution samples [26]. Nevertheless, these methods often show unstable results depending on the how hyperparameters are tuned [9, 20].

To overcome the instability and tuning complexity of the offline RL methods, bootstrapping-free methods that directly predict the future trajectories are also presented in recent studies [3, 20, 21, 1]. These methods, called sequence modeling methods, utilize the generative sequence models to predict future states and actions conditioned on the goal state or the sum of rewards along the trajectory. In particular, Janner et al. [21] and Ajay et al. [1] propose the methods to predict future trajectories using the denosing diffusion probabilistic models (DDPM), which have strengths in stitching the task relevant sub-trajectories from offline data and generating feasible trajectories under various

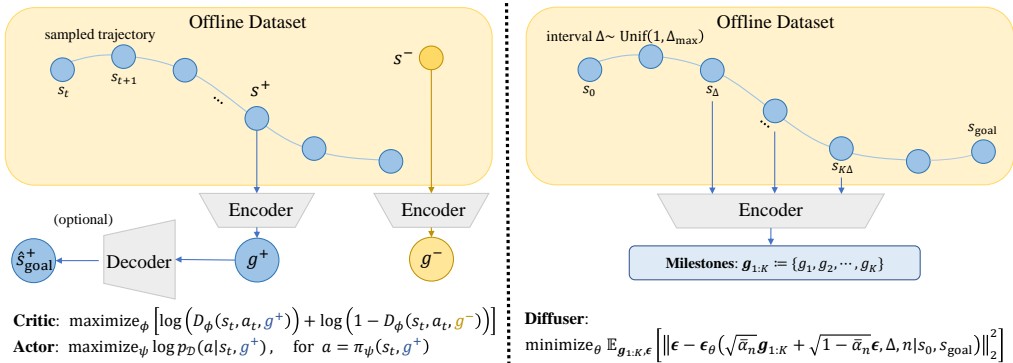

Figure 1: **Training process of DTAMP**. (Left) DTAMP learns the latent milestones through a goal-conditioned imitation learning manner. (Right) The diffusion model is trained to reconstruct the sequences of milestones sampled from the trajectories in the offline data.

combinations of constraints (*e.g.*, maximizing rewards, approaching to a given goal state). As a result, DDPM based models are able to flexibly plan trajectories and show promising results on multi-task learning. However, the inherent high computational cost of the denoising process limits its application to wider areas, such as real-time control and vision-based tasks.

In this paper, we present a method named **D**iffused **T**ask-**A**gnostic **M**ilestone **P**lanner (**DTAMP**) which aims to generate a sequence of *milestones* using the diffusion model to have an agent to accomplish a given task by following them. By doing so, DTAMP divides a long-horizon problem into shorter goal-reaching problems, and address them through goal-conditioned imitation learning which does not require the unstable bootstrapping method and has minimal hyperparameters to be tuned. Here, we note that a milestone represents a latent vector encoding a state (or an image), which guides an agent to reach the corresponding state. To learn the meaningful milestone representations, we propose a method to train an encoder along with a goal-conditioned actor and critic (Figure 1, Left). By doing so, we expect the encoder to extract the control-relevant features and to embed them into milestones. Then, we present a method to train the milestone planner to predict milestones for a given goal. To this end, we first sample a series of intermediate states at intervals from the trajectory approaching the goal, and encode them into milestones to train a diffusion model to reconstruct them. (Figure 1, Right). Furthermore, we also propose a method to guarantee that the milestone planner predicts the shortest path to the goal, based on the classifier-free diffusion guidance technique [17].

The main strengths of DTAMP are summarized as follows: 1) The encoder trained through the proposed method encodes control-relevant features into compact latent representations, which enables efficient planning even if the observations are high-dimensional. 2) The diffusion-based planner enables flexible planning for a variety of tasks and takes advantage on multi-task decision-making problems. 3) Using the trained actor as a feedback controller makes an agent robust against the environment stochasticity and makes it possible to plan milestones only once at the beginning of an episode which largely reduces inference time compared to the existing sequence modeling methods. The proposed method is first evaluated on the goal-oriented environments in the D4RL benchmark [8]. Especially, the results on Antmaze environments show that DTAMP outperforms offline RL methods in long-horizon tasks without using the bootstrapping method. Finally, DTAMP is demonstrated on the CALVIN benchmark [29], which is the most challenging image-based, multi-task environment, and shows the state-of-the-art performance achieving 1.7 times higher success rate compared to the second-best method. The results show that our approach can broaden the area of application of sequence modeling, while being superior to offline RL methods.

## 2 Preliminaries

### 2.1 Goal-conditioned imitation learning

Multi-task decision-making problems are often treated as a goal-reaching problem, assuming that each task can be specified by a goal state [28, 41, 35]. Accordingly, we consider the goal-reaching

problem in our work and handle the problem through a goal-conditioned imitation learning manner. We first formulate the problem with a goal-conditioned Markov decision process defined by a state space $\mathcal{S}$, an action space $\mathcal{A}$, a transition model $p(s'|s, a)$, an initial state distribution $\rho_0$, and a goal space $\mathcal{G}$. We assume that there exists a mapping $f : \mathcal{S} \to \mathcal{G}$, to determine whether the current state $s \in \mathcal{S}$ has reached a given goal $g \in \mathcal{G}$ by determining whether $f(s)$ equals to $g$. We further assume that there exists a set of previously collected data $\mathcal{D}$ which consists of the trajectories sampled by a behavior policy $\mu(a|s)$. Then, we define goal-conditioned imitation learning by training a goal-conditioned actor $\pi_\psi : \mathcal{S} \times \mathcal{G} \to \mathcal{A}$ to predict the most likely action to take for the goal $g \in \mathcal{G}$:

$$\text{maximize}_\psi \ \log \mu(a|s, g), \text{ for } s \in \mathcal{S} \text{ and } a = \pi_\psi(s, g). \tag{1}$$

We note that $\mu(a|s, g) = \frac{p_\mu(g|s,a)\mu(a|s)}{p_\mu(g|s)}$, where $p_\mu(g|s, a)$ and $p_\mu(g|s)$ are the finite horizon occupancy measures defined by,

$$p_\mu(g|s, a) := \sum_{s^+ \in \{s \in \mathcal{S} | f(s) = g\}} p_\mu(s^+|s, a) \text{ and } p_\mu(g|s) := \sum_{a \in \mathcal{A}} p_\mu(g|s, a)\mu(a|s),$$

$$\text{where, } p_\mu(s^+|s, a) = \mathbb{E}_{\mu(a_t|s_t), p(s_{t+1}|s_t, a_t)} \left[ \frac{1}{T} \sum_{t=1}^{T} P(s_t = s^+|s_0 = s, a_0 = a) \right]. \tag{2}$$

Then the objective in (1) can be reformulated as a loss function defined by,

$$J(s, g; \psi) := -\log p_\mu(g|s, a) - \log \mu(a|s), \text{ for } a = \pi_\psi(s, g). \tag{3}$$

We further note that the presented framework is highly related to the existing offline RL methods. Intuitively, the first term $\log p_\mu(g|s, a)$ in (15) is similar to the $Q$-functions used in the studies that address goal-conditioned RL problems [6, 7], and the second term $\log p_\mu(a|s)$ works as a regularization to prevent the policy from taking out-of-distribution actions [32, 11, 25].

## 2.2 Denosing diffusion probabilistic models

A denoising diffusion probabilistic model (DDPM) [18] is a generative model which learns to reconstruct the data distribution $q(\mathbf{x})$ by iteratively denoising noises sampled from $\mathcal{N}(\mathbf{0}, \mathbf{I})$. The diffusion models consist of the *diffusion process* which iteratively corrupts the original data $\mathbf{x} \sim q(\mathbf{x})$ into a white noise, and the *denoising process* which reconstructs the data distribution $q(\mathbf{x})$ from the corrupted data. First, in the diffusion process, a sequence of corrupting data $\mathbf{x}^{1:N} = \{\mathbf{x}^1, \mathbf{x}^2, \cdots, \mathbf{x}^N\}$ is sampled from a Markov chain defined by $q(\mathbf{x}^{n+1}|\mathbf{x}^n) = \mathcal{N}(\sqrt{1 - \beta_n}\mathbf{x}^n, \beta_n \mathbf{I})$, where $\beta_n \in (0, 1)$ is a variance schedule and $n = 0, 1, \cdots, N$ indicates each diffusion timestep. Then the denoising process aims to learn a denoising model $p_\theta(\mathbf{x}^{n-1}|\mathbf{x}^n)$, to sequentially denoise the noisy data $\boldsymbol{x}^N \sim \mathcal{N}(\mathbf{0}, \boldsymbol{I})$ into the original data $\boldsymbol{x}^0$.

The denoising model $p_\theta(\mathbf{x}^{n-1}|\mathbf{x}^n)$ can be trained by minimizing the following variational bound:

$$\mathbb{E}_q \left[ D_{\text{KL}}\big(q(\mathbf{x}_N|\mathbf{x}_0)\|p(\mathbf{x}_N)\big) + \sum_{n=2}^{N} D_{\text{KL}}\big(q(\mathbf{x}^{n-1}|\mathbf{x}^n, \mathbf{x}^0)\|p_\theta(\mathbf{x}^{n-1}|\mathbf{x}^n)\big) - \log p_\theta(\mathbf{x}^0|\mathbf{x}^1) \right]. \tag{4}$$

However, the variational bound (4) is often reformulated to a simpler rescaled loss function using a function approximator $\boldsymbol{\epsilon}_\theta$ to predict the noise injected into the original data.

$$J(\mathbf{x}^0, n; \theta) := \mathbb{E}_{\boldsymbol{\epsilon} \sim \mathcal{N}(\mathbf{0}, \mathbf{I})} \left[ \|\boldsymbol{\epsilon} - \boldsymbol{\epsilon}_\theta(\sqrt{\bar{\alpha}_n}\mathbf{x}^0 + \sqrt{1 - \bar{\alpha}_n}\boldsymbol{\epsilon}, n)\|^2 \right], \text{ where } \bar{\alpha}_n = \prod_{m=0}^{n} (1 - \beta_m). \tag{5}$$

The surrogate loss function (5) is known to reduce the variance of the loss and improve the training stability [18].

## 3 Diffused task-agnostic milestone planner

In this section, we introduce diffused task-agnostic milestone planner (DTAMP). We first present the goal-conditioned imitation learning method for simultaneously training the encoder, the actor, and the critic functions. Secondly, the method to train the diffusion model to plan milestones is presented. In addition, we also propose a diffusion guidance method to let the milestone planner predict the shortest path to reach a given goal. Finally, we present a sequential decision-making algorithm combining the goal-conditioned actor and the milestone planner. Note that more detailed explanation about our implementation of DTAMP is presented in Appendix D.

### 3.1 Goal-conditioned imitation learning from offline data

We first introduce our approach to train the goal-conditioned actor $\pi_\psi$, using a discriminative function $D_\phi : \mathcal{S} \times \mathcal{A} \times \mathcal{G} \to [0, 1]$ as the critic function to estimate the log-likelihood $\log p_\mu(g|s, a)$ in (15). Here, we assume that a goal is designated by a single state $s_{\text{goal}} \in \mathcal{S}$ which can be a raw sensory observation or an image. In order to extract the underlying goal representation $g \in \mathcal{G}$ from the given goal state, we also train the encoder $f_\omega : \mathcal{S} \to \mathcal{G}$ alongside the actor and critic, where $\mathcal{G}$ is a latent goal space. Note that the more discussion about how the encoder is trained through imitation learning and visualization of the learned goal space are presented in Appendix G.

To estimate the log-likelihood $\log p_\mu(g|s, a)$ in (15), we first train the discriminative critic $D_\phi$ to minimize the following cross-entropy loss:

$$J_{\text{critic}}(s, a, s^+, s^-; \phi, \omega) := -\log D_\phi(s, a, f_\omega(s^+)) - \log (1 - D_\phi(s, a, f_\omega(s^-))),$$
$$\text{where, } s \sim p_\mu(s), \; a \sim \mu(a|s), \; s^+ \sim p_\mu(s^+|s, a), \; s^- \sim p_\mu(s^-). \tag{6}$$

We note that $p_\mu(s)$ is a prior distribution over the state space computed by $\sum_{s_0 \in \mathcal{S}} p_\mu(s|s_0)\rho_0(s_0)$, which is estimated by a uniform distribution over the all data points of the offline data. Intuitively, for a given state-action pair $(s, a)$, the positive sample $s^+$ is sampled from the future states comes after $(s, a)$, and the negative sample $s^-$ is sampled uniformly across the entire data distribution. As a result, we expect the critic to be trained to distinguish between states that are more likely to come after the current state-action pair. This intuition can be formally formulated with the following proposition.

**Proposition 3.1.** *The optimal critic function $D_\phi^*$ that minimizes the cross-entropy loss (16) satisfies the following equation for a given goal $g = f_\omega(s_{\text{goal}})$.*

$$D_\phi^*(s, a, g) = \frac{p_\mu(g|s, a)}{p_\mu(g|s, a) + p_\mu(g)}. \tag{7}$$

*Proof.* Proof of Proposition A.1 is presented in Appendix A. □

Using Proposition A.1, we estimate $p_\mu(g|s, a)$ for a given goal $g \in \mathcal{G}$ with the trained critic function.

$$p_\mu(g|s, a) = p_\mu(g) \frac{D_\phi(s, a, g)}{1 - D_\phi(s, a, g)}. \tag{8}$$

Now, the remaining part is to estimate $\log p_\mu(a|s)$ in (15) to train the policy $\pi_\psi$. However, although some previous studies try to estimate the marginal likelihood using conditional generative models [11] or density estimation methods [47, 24], they require additional computational cost while not showing the significant improvement in performance [9]. Therefore, we circumvent this issue by replacing $\log p_\mu(a|s)$ in the objective (15) into a simple behavior-cloning regularization term. This regularization has been shown to be an effective way to stabilize offline policy learning requiring minimal hyperparameter tuning [9]. Consequently, the training objective for the policy function is defined as the following loss function:

$$J_{\text{actor}}(s, \tilde{a}, s^+; \psi, \omega) := -\lambda \log \frac{D_\phi(s, \tilde{a}, f_\omega(s^+))}{1 - D_\phi(s, \tilde{a}, f_\omega(s^+))} + \|\tilde{a} - a\|^2,$$
$$\text{where, } s \sim p_\mu(s), \; a \sim \mu(a|s), \; s^+ \sim p_\mu(s^+|s, a), \; \tilde{a} = \pi_\psi(s, f_\omega(s^+)). \tag{9}$$

where $\lambda$ is a hyperparameter which balances the weight between the maximizing log-likelihood $\log p_\mu(f_\omega(s^+)|s, \tilde{a})$ and the regularization.

Please note that training the actor and the critic through the proposed goal-conditioned imitation learning method does not require temporal difference learning (or bootstrapping). Meanwhile, as the actor is trained only for the positive goals, it has its limitation in inferring a suitable action for a negative goal that has never been reached from the current state in the data collection process. To overcome this limitation, we further propose a diffusion-based milestone planner which predicts milestones to guide an agent to reach the distant negative goal state, in the next section.

---

| Algorithm 1: Sequential decision-making using DTAMP |
| --- |

1: Predict milestones $\boldsymbol{g}_{1:K} \sim p_\theta(\boldsymbol{g}_{1:K}^0|\boldsymbol{g}_{1:K}^N, \triangle_{\text{target}}, s_0, s_{\text{goal}})$, through denoising process (11).
2: Set the index $k$ for targeting a milestone: $k \leftarrow 1$.
3: Set internal timestep $\tau \leftarrow 0$.
4: **for** each environment timestep $t$ **do**
5:     Take an environment step with an action $a = \pi_\psi(s_t, g_k)$.
6:     Increase internal timestep $\tau \leftarrow \tau + 1$.
7:     **if** $\|f_\omega(s_{t+1}) - g_k\|_2^2 < \delta$ or $\tau > \tau_{\text{lim}}$ **then**
8:         Switch the target milestone to the next: $k \leftarrow k + 1$.
9:         Reset internal timestep $\tau \leftarrow 0$.
10:    **end if**
11: **end for**

---

## 3.2 Diffusion model as a milestone planner

We now introduce our method to train the diffusion model as a milestone planner to guide the actor to reach the distant goal state. Intuitively, this can be done by letting the diffusion model to predict intermediate sub-goals that should pass. To this end, we first sample from offline data a pair of initial and goal state $(s_0, s_{\text{goal}})$ and $K$-intermediate states $\{s_\triangle, \cdots, s_{K\triangle}\}$ at interval $\triangle$, where $s_{(K+1)\triangle} = s_{\text{goal}}$. Then the sampled intermediate states are encoded into a series of latent milestones $\boldsymbol{g}_{1:K} := \{g_1, g_2, \cdots, g_K\}$, such that $g_k = f_\omega(s_{k\triangle})$ for $k = 1, 2, \cdots, K$. Then, the diffusion model is trained to reconstruct $\boldsymbol{g}_{1:K}$ by minimizing the following loss function:

$$J_{\text{diffusion}}(\boldsymbol{g}_{1:K}, n, s_0, s_{\text{goal}}; \theta) := \mathbb{E}_{\boldsymbol{\epsilon} \sim \mathcal{N}(\mathbf{0},\mathbf{I})} \left[ \|\boldsymbol{\epsilon} - \boldsymbol{\epsilon}_\theta(\sqrt{\bar{\alpha}_n}\boldsymbol{g}_{1:K} + \sqrt{1 - \bar{\alpha}_n}\boldsymbol{\epsilon}, n|s_0, s_{\text{goal}})\|^2 \right]. \tag{10}$$

Then, the trained milestone planner can predict milestones for a given pair of initial and goal state $(s_0, s_{\text{goal}})$ by sequentially denoising a series of noisy milestones $\boldsymbol{g}_{1:K}^N \sim \mathcal{N}(\mathbf{0}, \boldsymbol{I})$:

$$\boldsymbol{g}_{1:K}^{n-1} \sim \mathcal{N}\left( \sqrt{\frac{\bar{\alpha}_{n-1}}{\bar{\alpha}_n}}(\boldsymbol{g}_{1:K}^n - \sqrt{1 - \bar{\alpha}_n}\hat{\boldsymbol{\epsilon}}_n), \sqrt{1 - \bar{\alpha}_{n-1}}\boldsymbol{I} \right), \tag{11}$$

for $\hat{\epsilon}_n = \boldsymbol{\epsilon}_\theta\left(\boldsymbol{g}_{1:K}^n, n|s_0, s_{\text{goal}}\right)$ and $n = N, N-1, \cdots, 1$.

## 3.3 Minimum temporal distance diffusion guidance

The denoising process (11) makes it possible to generate milestones for the given goal state, but does not guarantee that the planned trajectory is the shortest path. To address this issue we make an additional modification using classifier-free diffusion guidance [17]. Since we are interested in minimizing the length of trajectories represented by milestones, minimizing the temporal distances between the pairs of successive milestones is a natural choice. Therefore, we modify the diffusion model to be conditioned on the temporal interval $\triangle$ of the sampled sequence of intermediate states $\{s_\triangle, \cdots, s_{K\triangle}\}$. The loss function (5) is then extended to the following unconditional loss function $J_{\text{uncond}}$ and conditional loss function $J_{\text{cond}}$:

$$J_{\text{uncond}}(\boldsymbol{g}_{1:K}, n; \theta) = \mathbb{E}_{\boldsymbol{\epsilon} \sim \mathcal{N}(\mathbf{0},\mathbf{I})} \left[ \|\boldsymbol{\epsilon} - \boldsymbol{\epsilon}_\theta(\sqrt{\bar{\alpha}_n}\boldsymbol{g}_{1:K} + \sqrt{1 - \bar{\alpha}_n}\boldsymbol{\epsilon}, \varnothing, n)\|^2 \right],$$
$$J_{\text{cond}}(\boldsymbol{g}_{1:K}, \triangle, n; \theta) = \mathbb{E}_{\boldsymbol{\epsilon} \sim \mathcal{N}(\mathbf{0},\mathbf{I})} \left[ \|\boldsymbol{\epsilon} - \boldsymbol{\epsilon}_\theta(\sqrt{\bar{\alpha}_n}\boldsymbol{g}_{1:K} + \sqrt{1 - \bar{\alpha}_n}\boldsymbol{\epsilon}, \triangle, n)\|^2 \right]. \tag{12}$$

Then, the shortest path planning is done with the guided prediction of $\hat{\epsilon}_n$ in (11):

$$\hat{\epsilon}_n = \boldsymbol{\epsilon}_\theta(\boldsymbol{g}_{0:K}^n, \varnothing, n) + \beta \left[ \boldsymbol{\epsilon}_\theta(\boldsymbol{g}_{0:K}^n, \triangle_{\text{target}}, n) - \boldsymbol{\epsilon}_\theta(\boldsymbol{g}_{0:K}^n, \varnothing, n) \right], \tag{13}$$

where $\triangle_{\text{target}}$ denotes the target temporal distance between milestones which is set to be relatively smaller than the maximum value $\triangle_{\text{max}}$, and $\beta$ is a scalar value that determines the weight of the guidance. Please note that the conditions on the pair of initial and goal state $(s_0, s_{\text{goal}})$ are omitted in (12) and (13) for the legibility.

## 3.4 Sequential decision-making using DTAMP

Using the learned milestone planner, we can predict a series of milestones for a given goal state, and let the agent to follow the milestones using the learned actor function. We note that the goal-conditioned actor works as a feedback controller which can adapt to stochastic transitions. This

| | Environment | CQL | IQL | ContRL | DT | TT(+IQL) | DD | **DTAMP** |
|---|---|---|---|---|---|---|---|---|
| antmaze | medium-play | 61.2 | 71.2 | 72.6 | 0.0 | 0.0 (81.9) | 0.0 | **93.3**±0.94 |
| | medium-diverse | 53.7 | 70.0 | 71.5 | 0.0 | 0.0 (85.7) | 0.0 | **88.7**±3.86 |
| | large-play | 15.8 | 39.6 | 48.6 | 0.0 | 0.0 (64.8) | 0.0 | **80.0**±3.27 |
| | large-diverse | 14.9 | 47.5 | 54.1 | 0.0 | 0.0 (65.7) | 0.0 | **78.0**±8.83 |
| | Average | 36.4 | 57.1 | 61.7 | 0.0 | 0.0 (74.5) | 0.0 | **85.0**±4.84 |
| kitchen | mixed | 52.4 | 51.0 | - | - | - | 65.0 | **74.4**±1.39 |
| | partial | 50.1 | 46.3 | - | - | - | 57.0 | **63.4**±8.80 |
| | Average | 51.3 | 48.7 | - | - | - | 61.0 | **68.9**±6.30 |

Table 1: The table shows averaged scores on D4RL benchmarks. DTAMP was trained with three different random seeds, and evaluated with 100 rollouts for each random seed. The source of the baseline performance is presented in Appendix C.

property allows to perform the time-consuming denoising process only once at the beginning of an episode. In contrast, the other sequence modeling methods [20, 21, 1] have to predict future trajectories at every timestep, which largely increases inference time. The remaining important question is how to decide whether the agent has reached the current milestone and move on to the next one. We address this issue by setting a threshold $\delta$ and make the switch if the distance between the current state and the targeted milestone measured in the latent space is less then the threshold. In addition, to make the agent recover from failure to reach a milestone, we also set an internal time limit $\tau_{lim}$ and allow the agent to move on to the next milestone when it fails to reach the current one within the time limit. Algorithm 1 summarizes our method for sequential decision-making.

## 4 Experiments

In this section we empirically verify that DTAMP is able to handle long-horizon, multi-task decision-making problems. We first evaluate the proposed method on the goal-oriented environments in the D4RL benchmark [8], and show that our approach outperforms the other methods on the long-horizon, sparse-reward tasks. Then, we evaluate DTAMP on the most challenging CALVIN benchmark [29], and show that DTAMP achieves the state-of-the-art performance. Finally, we also present an ablation study to investigate how DTAMP's performance is influenced by the minimum time distance guidance method (13) and the various other design choices of DTAMP. Please note that the detailed experiment settings are presented in Appendix B, and additional experiments on the D4RL locomotion tasks and a stochastic environment are presented in Appendix H.

### 4.1 Offline reinforcement learning

**D4RL benchmarks**   We first evaluate DTAMP on the goal-oriented environments in the D4RL benchmark (Antmaze and Kitchen environments) to verify that DTAMP can handle long-horizon tasks without bootstrapping the value function. These tasks are known to be especially challenging to be solved as the useful reward signals only appear when the agent reaches the goal states that are far apart from starting states. The proposed method is compared against offline RL baselines (**CQL**: conservative q-learning [26], **IQL**: implicit q-learning [25], **ContRL**: contrastive reinforcement learning [7]), and the sequence modeling methods (**DT**: decision transformer [3], **TT**: trajectory transformer [20], **DD**: decision diffuser [1]). In addition, we also compare DTAMP to a variant of TT which utilizes a value function trained by IQL for using the beam search algorithm (**TT+IQL**), as proposed by Janner et al. [20].

The results in Table 1 show that DTAMP outperforms the baseline methods on the every environment. Especially, the other sequence modeling methods (DT, TT, and DD) that do not use temporal difference learning fail to show any meaningful result on the Antmaze environments. The reason for their poor performance is that the goal state of the Antmaze environment is too far from the initial

| | Environment | IQL+HER | ContRL | TT+IQL | **DTAMP** |
|---|---|---|---|---|---|
| antmaze multi-goal | medium-play | 34.7 (-51.3%) | 58.7 (-19.1%) | 68.9 (-15.9%) | **89.3**±3.86 (-4.3%) |
| | medium-diverse | 50.7 (-27.6%) | 57.3 (-19.9%) | 75.6 (-11.8%) | **76.7**±4.50 (-13.5%) |
| | large-play | 46.0 (+16.2%) | 24.7 (-49.2%) | 28.9 (-55.4%) | **62.0**±2.94 (-22.5%) |
| | large-diverse | 42.0 (-11.6%) | 26.7 (-50.6%) | 23.3 (-64.5%) | **53.3**±9.67 (-31.7%) |
| | Average | 43.4 (-24.0%) | 41.9 (-32.2%) | 49.2 (-34.0%) | **70.3**±5.86 (-17.3%) |

Table 2: The table shows averaged scores on the multi-goal setting of Antmaze environments. The values in the parenthesises indicate performance difference between single-goal and multi-goal settings in percentage.

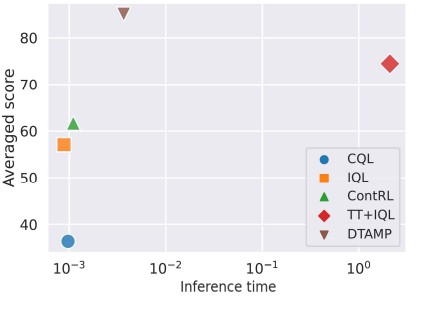

(a) Antmaze environments

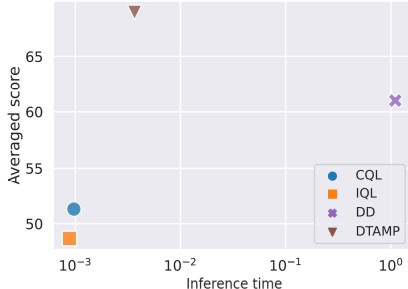

(b) Kitchen environments

Figure 2: The plots compare the inference times and the average scores of the different algorithms on the D4RL benchmarks. The $x$-axis indicates the inference time in logarithmic scale. The inference times are measured using an NVIDIA GeForce 3060 Ti graphics card.

state[1] to accurately predict the whole trajectory to the goal. Meanwhile, DTAMP can efficiently plan a long-horizon trajectory by generating temporally sparse milestones.

**Multi-goal Antmaze experiment**   In addition, we demonstrate DTAMP on Antmaze environments with multi-goal setting to show that DTAMP can flexibly plan the milestones according to the varying goals. In this experiment, a goal for each rollout is randomly sampled from three different predefined target positions including the goals that were not reached from the initial state during the data-collection (see Appendix B for more details). We compare DTAMP against the three baseline methods (**IQL** [25], **ContRL** [7], and **TT+IQL** [20]), which achieve high scores in the single-goal setting. We note that DTAMP and ContRL already learn goal-conditioned actors and do not require to be modified for the multi-goal setting. Meanwhile, we modify IQL and TT+IQL by adding hindsight experience replay (HER) [2]. In the multi-goal setting, DTAMP shows the least performance degradation compared to the other methods (Table 2). Especially, TT+IQL shows the largest performance degradation compared to the results evaluated on the single-goal setting. This result indicates that the transformer architecture [45] used in TT struggles to predict the trajectory to reach a new goal that was not seen in the training time, while DTAMP adaptively plan the milestones by stitching the relevant sub-trajectories.

**Comparison of inference time**   In order to evaluate the computational efficiency of the proposed method, we further compare DTAMP against the other methods in terms of inference time it takes to predict a single step of action. The comparative result in Figure 2 shows that our approach largely reduces the computational cost compared to the other sequence modeling methods, while achieving the highest average score. In particular, the inference time of DTAMP is about 1,000 times faster than TT and about 300 times faster than DD. It is because using the actor as a feedback controller makes the agent robust on the stochasticity of the environment, and allows to perform the time-consuming sequence generation process only once at the beginning of an episode. In contrast, the other sequence

---

[1]about 500 timesteps apart

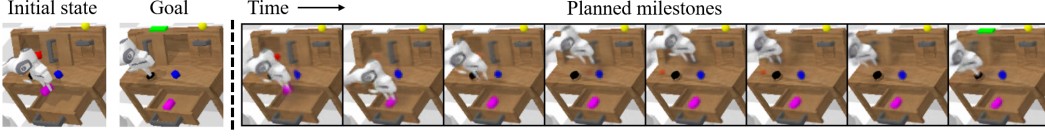

Initial state  Goal  Time ⟶  Planned milestones

Implied tasks: {'place pink block in drawer', 'move slider left', 'turn on led'}

Figure 3: Images on the left show a pair of initial state and goal state, which implies three different tasks. The series of images on the right represents the milestones planned by DTAMP. The image of each milestone is reconstructed using the trained decoder.

modeling methods have to predict the future trajectories for every timestep which results in high computational cost.

## 4.2 Image-based planning

We now evaluate the proposed method on the CALVIN benchmark [29], which is the most challenging multi-task visual manipulation task. In this experiment, the observations and goals are designated by images, and reaching a goal image may requires the agent to sequentially accomplish multiple tasks. Figure 3 shows an example that three different tasks (`place pink block in drawer`, `move slider left`, and `turn on led`) are implied by a single goal image. We utilize the dataset provided by Mees et al. [29], which consists of the trajectories performing a variety of manipulation tasks (*e.g.*, picking up a block, opening the drawer, pushing the button to turn on the led, etc.) collected by a human teleoperating the robotic arm in the simulation. To investigate how the performance of DTAMP and other baselines varies with the difficulty of the task, we group the pairs of initial and goal states into three groups according to the number of tasks they imply, and evaluate the models on each group separately.

We note that two modifications are added to DTAMP in this experiment, as the CALVIN environment has the partially observable characteristic. First, we additionally train a decoder to reconstruct image observations from the encoded latent milestone vectors in order to let the encoder capture the essential visual features. Second, we utilize skill-based policy as done by Lynch et al. [28] and Rosete-Beas et al. [35], to let the agent predict actions based on the current and previous observations. More detailed discussion of how modifications affect the performance of DTAMP will be presented in the ablation study.

In this experiment, our method is compared against an offline RL baseline combined with goal rela-belling (**CQL+HER**: conservative q-learning [26] with hindsight experience replay [2]) and hierarchi-cal RL methods (**LMP**: latent motor plans [28], **RIL**: relay imitation learning [13], **TACO-RL**: task-agnostic offline reinforcement learning [35]). Furthermore, we also evaluate **DTAMP+Replanning** which allows the agent to re-plan the rest of milestones at every time it reaches a milestone. The results in Table 3 shows that our approach achieves the state-of-the-art performance in the CALVIN environment against existing methods by a significant margin. We also would like note that only DTAMP and DTAMP+Replanning show the meaningful success rates on the hardest setting that a goal image implies three different tasks in a row. The results indicate that our approach enables planning on image-based environments, while also outperforming the other hierarchical RL methods that utilize skill-based policy (LMP and TACO-RL) or generate sub-goals (RIL). Furthermore, we also see that the performance of DTAMP can be further improved by allowing agents to re-plan milestones during an episode.

## 4.3 Ablation study

**Minimum temporal distance guidance** We present an ablation study to verify that the minimum temporal distance guidance (13) enables DTAMP to plan the shortest path for a given goal. We demonstrate DTAMP on a simplified open Antmaze environment with varying target temporal distance $\triangle_{\text{target}}$. In order to train DTAMP, one million steps of offline data was collected by a pretrained locomotion policy. Then, the unconditional version (10) and conditional version (12) of DTAMP were trained using the collected data. The trajectories planned by the unconditional model and conditional model are show in Figure 4. As expected, the guidance method enables DTAMP to plan trajectories of varying lengths according to the given target temporal distances, while the

| Number of tasks | CQL+HER | LMP | RIL | TACO-RL | **DTAMP** | **DTAMP+Replanning** |
|---|---|---|---|---|---|---|
| 1 | 58.0 | 61.4 | 66.2 | 80.3 | 82.4±2.87 | **87.4**±1.43 |
| 2 | 2.4 | 2.7 | 13.3 | 27.0 | 52.4±1.64 | **62.6**±3.57 |
| 3 | 0.0 | 0.0 | 0.0 | 0.4 | 29.7±6.80 | **38.1**±5.78 |
| Average | 20.1 | 21.3 | 26.5 | 35.9 | 54.8±4.37 | **62.7**±3.60 |

Table 3: The table shows averaged success rates on the CALVIN benchmark in percentage. DTAMP is trained with three random seeds, and the each model is evaluated with 1000 rollouts.

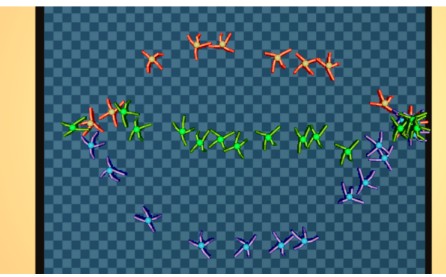

| $\triangle_{\text{target}}/\triangle_{\text{max}}$ | Success rate (%) | Timesteps to goal |
|---|---|---|
| 0.1 | 97.7 ±1.77 | 178 ±3.46 |
| 1.0 | 96.3 ±1.89 | 310 ±10.9 |
| Uncond | 96.7 ±1.70 | 249 ±18.8 |

Figure 4: The figure visualizes the milestones planned under different conditions (red: unconditioned, green: $\triangle_{\text{target}}/\triangle_{\text{max}} = 0.1$, blue: $\triangle_{\text{target}}/\triangle_{\text{max}} = 1.0$)

Table 4: The table shows averaged success rates and the number of timesteps taken to reach the goal under different conditions.

unconditional model tends to plan sub-optimal trajectories. In addition, Table 4 shows that the guidance method is able to reduce the number of timesteps required to reach the goal, while it does not affect the success rate.

**Ablation study in image-based planning** In order to apply DTAMP on the image-based planning problems, we made two modifications on DTAMP: 1) Training a decoder to reconstruct images. 2) Utilizing skill-based policy. In this experiment we further investigate how each modification affects to the performance of DTAMP. To this end, we evaluate three variations of DTAMP in the CALVIN benchmark (**DTAMP−Recon. Loss**: the encoder is trained without reconstruction loss, **DTAMP−GCIL Loss**: the encoder is trained with only reconstruction loss without goal-conditioned imitation learning loss, **DTAMP−LMP**: the agent does not utilize skill-based policy). The results shown in Table 5 indicate that using both reconstruction loss and GCIL loss when training the encoder improves DTAMP's performance the most. We also find that ablating skill-based policy causes a large performance degradation.

## 5 Related Work

**Diffusion models** [39, 18, 38] have received active attention recently. There exists a number of applications of diffusion model, such as photo-realistic image generation [5, 34], text-guided image generation [36, 31, 33], and video generation [19, 15, 16, 50]. While the majority of work related to diffusion model is focused on image generation and manipulation, there is also growing interest in applying diffusion model on robotic tasks or reinforcement learning. The methods to plan a trajectory with diffusion models to perform a given task are proposed by Janner et al. [21] and Ajay et al. [1]. In the meantime, Wang et al. [46] and Chi et al. [4] present the methods to utilize diffusion models for predicting actions. Furthermore, Urain et al. [42] proposes a method to train a cost function alongside with a diffusion model, and to utilize it to select a proper grasping angle. It is worth mentioning that our approach is the first work that utilizes diffusion model to plan trajectories in a learnable latent space and handle visuomotor tasks, to the best of our knowledge.

**Offline reinforcement learning** has also been widely studied to improve data efficiency of reinforcement learning (RL) algorithms [30, 37, 44, 27, 14]. The most of RL methods utilize temporal difference method to estimate the value of current action considering over the further timesteps. How-

| Number of tasks | DTAMP | DTAMP −Recon. Loss | DTAMP −GCIL Loss | DTAMP − LMP |
|---|---|---|---|---|
| 1 | 82.4 | 80.2 | **86.7** | 55.9 |
| 2 | **52.4** | 49.4 | 43.3 | 24.3 |
| 3 | **29.7** | 23.0 | 17.2 | 15.2 |
| Average | **54.8** | 50.9 | 49.1 | 31.8 |

Table 5: The results of ablation study on the CALVIN benchmark. The values represent success rates in percentage.

ever, bootstrapping the estimated values often causes overestimating the value of out-of-distribution actions [10, 11], which makes it challenging to apply RL on fixed offline data. To tackle this issue, Fujimoto et al. [11] proposes batch constrained policy learning which restricts the policy to output the actions within the data. There also various methods to stabilize the offline RL are proposed, by learning policy through advantage weighted regression [32], augmenting q-learning objective [26, 25], or model-based reinforcement learning [22]. In addition, applying offline RL on multi-task RL problems is showing promising results in recent years [35], as it is able to extract various skills from offline data and compose them to accomplish a new challenging task. Accordingly, our work also considers the offline setting that leverages the previously collected data, and mainly focus on designing a method to stitch the separated trajectories in the offline data to accomplich a given task.

## 6 Limitations

**Cannot solve a new task not included in dataset**. In this work, we considered multi-task decision-making problems assuming that the offline data contains the segments of trajectories for performing the tasks we are interested in. However, there may also be cases where an agent needs to find new skills to solve new tasks that were not included in the previously collected data. Exploring the environment using the pretrained model to find new skills and continually learning to perform more challenging tasks would be an interesting topic for the future work.

**Efficient re-planning strategy is needed**. Although the proposed method leverages faster inference time compared to other generative model-based planning methods by allowing the agent to plan milestones only once, our empirical results show that having the agent re-plan the trajectories during episodes can improve performance. Furthermore, in order to apply DTAMP on the problems that their environments are constantly changing, the agent is required to re-plan the trajectories according to the changed environment. In this context, an efficient re-planning strategy to determine when and how to modify the planned path must be further studied to utilize DTAMP for a wider area of robotic tasks.

## 7 Conclusion

This paper presents a method to plan a sequence of latent sub-goals, which is named as milestones, for a given target state to be reached. The proposed method trains an encoder to extract milestone representations from observations, and an actor to predict actions to follow the milestones, through a goal-conditioned imitation learning manner. Our method also utilizes a denosing diffusion model to generate an array of milestones conditioned on a target state. Furthermore, the minimum temporal distance diffusion guidance method is also presented in this paper, to make the proposed method plan the shortest path to reach the target state. Experimental results support that the proposed method makes it possible to plan the milestones adapting to various tasks, and to manage long-term tasks without using the unstable bootstrapping method.

## Acknowledgements

This work was supported by Institute of Information & Communications Technology Planning & Evaluation (IITP) grant funded by the Korea government (MSIT) (No. 2019-0-01190, [SW Star Lab] Robot Learning: Efficient, Safe, and Socially-Acceptable Machine Learning).

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

# Appendix

## A   Proof of Proposition 1

**Recap**   We consider a goal-conditioned Markov decision process defined by a state space $\mathcal{S}$, an action space $\mathcal{A}$, a transition model $p(s'|s, a)$, and a goal space $\mathcal{G}$. We further consider an encoder $f_\omega : \mathcal{S} \to \mathcal{G}$ which maps a goal state $s_{\text{goal}} \in \mathcal{S}$ into a goal vector $g \in \mathcal{G}$. Goal-conditioned imitation learning is defined by training a goal-conditioned actor $\pi_\psi : \mathcal{S} \times \mathcal{G} \to \mathcal{A}$ to predict the most likely action that a behavior policy $\mu(a|s)$ will take for a given goal $g \in \mathcal{G}$:

$$\text{maximize}_\psi \ \log \mu(a|s, g), \ \text{for } s \in \mathcal{S} \text{ and } a = \pi_\psi(s, g). \tag{14}$$

Using Bayes' rule, $\mu(a|s, g) = \frac{p_\mu(g|s,a)\mu(a|s)}{p_\mu(g|s)}$, where $p_\mu(g|s, a)$ and $p_\mu(g|s)$ are the finite horizon occupancy measures, the objective in (14) can be reformulated as:

$$J(s, g; \psi) := -\log p_\mu(g|s, a) - \log \mu(a|s), \ \text{for } a = \pi_\psi(s, g). \tag{15}$$

In order to estimate the log-likelihood $\log p_\mu(g|s, a)$, we train a discriminative critic function $D_\phi : \mathcal{S} \times \mathcal{A} \times \mathcal{G} \to [0, 1]$ to distinguish between states that are more likely to come after the current state-action pair, by minimizing the following loss function:

$$J_{\text{critic}}(s, a, s^+, s^-; \phi, \omega) := -\log D_\phi(s, a, f_\omega(s^+)) - \log\left(1 - D_\phi(s, a, f_\omega(s^-))\right),$$
$$\text{where, } s \sim p_\mu(s), \ a \sim \mu(a|s), \ s^+ \sim p_\mu(s^+|s, a), \ s^- \sim p_\mu(s^-).\text{`} \tag{16}$$

We now aim to prove the following proposition:

**Proposition A.1.** *The optimal critic function $D_\phi^*$ that minimizes the cross-entropy loss (16) satisfies the following equation for a given goal goal $g = f_\omega(s_{goal})$.*

$$D_\phi^*(s, a, g) = \frac{p_\mu(g|s, a)}{p_\mu(g|s, a) + p_\mu(g)}. \tag{17}$$

*Proof.* We can rewrite the loss function (16) as,

$$J(s, a; \phi, \omega) = -\sum_{\tilde{g} \in \tilde{\mathcal{G}}_\omega} \left[p_\mu(\tilde{g}|s, a) \log D_\phi(s, a, \tilde{g}) + p_\mu(\tilde{g}) \log\left(1 - D_\phi(s, a, \tilde{g})\right)\right], \tag{18}$$

where, $\tilde{\mathcal{G}}_\omega = \{f_\omega(s)|s \in \mathcal{S}\}$. Then, by taking derivative of (18) with respect to $D_\phi(s, a, g)$,

$$\frac{\partial J(s, a; \phi, \omega)}{\partial D_\omega(s, a, g)} = -\frac{p_\mu(g|s, a)}{D_\omega(s, a, g)} + \frac{p_\mu(g)}{1 - D_\omega(s, a, g)}. \tag{19}$$

At $D_\phi(s, a, g) = D_\phi^*(s, a, g)$, the derivative in (19) becomes $0$. Therefore,

$$D_\phi^*(s, a, g) = \frac{p_\mu(g|s, a)}{p_\mu(g|s, a) + p_\mu(g)}.$$

$\square$

## B   Experiment setting

**Multi-goal Antmaze experiments**   To evaluate DTAMP and baseline methods on multi-goal setting, we predefined three different target positions as shown in Figure 5. During the evaluation, a goal was randomly sampled from the three target positions for each rollout. The models were trained using the same data (`antmaze-medium-play`, `antmaze-medium-diverse`, `antmaze-large-play`, `antmaze-large-diverse`) that was used for single-goal Antmaze experiments.

**CALVIN experiments**   We used the same data split and tasks used in Rosete-Beas et al. [35] to train and evaluate the model in the CALVIN experiment. We set time horizon of a rollout to 300 timesteps when a goal image implies one or two tasks, and 450 timesteps when a goal image implies three tasks.

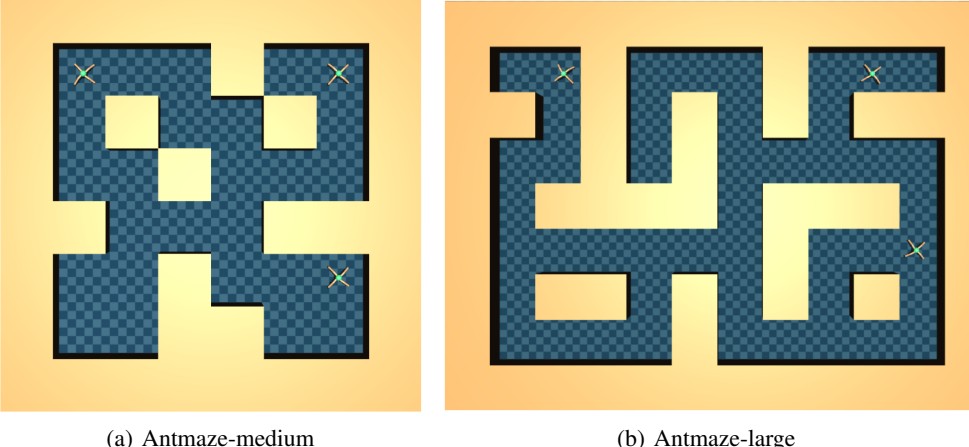

(a) Antmaze-medium          (b) Antmaze-large

Figure 5: Each position marked by an ant illustrates a predefined target position.

## C    Source of baseline performance

**D4RL experiments**    The performances of CQL [26], IQL [25], and ContRL [7] are taken from their corresponding papers. The performance of DT [3] in Antmaze environments is taken from Janner et al. [20]. The performance of DD [1] in Kitchen environment is taken from its corresponding paper. To evaluate DD on Antmaze environments, DD is trained with three different random seeds using the author's code[2], and each trained model is evaluated with 30 rollouts (90 rollouts in total) for each task. While the performance of TT+IQL is presented in Janner et al. [20], it is inaccurate as only 15 rollouts were used for evaluation. For more accurate evaluation, TT and IQL are trained with three different random seeds using the code[3] provided by the author, and each model is evaluated with 30 rollouts (90 rollouts in total) for each task.

**Multi-goal Antmaze experiments**    IQL+HER is implemented based on the author's code[4] by concatenating two-dimensional goal position on the original state vector. We train IQL+HER with three different random seeds and evaluate each model with 150 rollouts. As ContRL already learns a goal-conditioned policy, ContRL is trained with three different random seeds using the author's code[5] without modification. As also TT does not require a goal-specific training, we evaluate TT+IQL using the model trained for single-goal experiments while using the value functions trained by IQL+HER.

**CALVIN experiments**    The performances of CQL+HER, LMP [28], RIL [13], and TACO-RL [35] are taken from Rosete-Beas et al. [35] for the cases that one or two tasks are implied by a goal image. For the case that three tasks are implied by a goal image, we trained each model using the code[6] provided by Rosete-Beas et al. [35], and evaluated each model with 1,000 rollouts.

## D    Implementation details

**Encoder**    The encoder consists of two neural networks one for each actor and critic. Each neural network has two hidden fully-connected layers with a size of 512, and an output layer. In addition, we normalized each output of the two neural networks, to provide consistent signal-to-noise ratio to the diffusion model during training process.

**Actor and critic**    Each actor and critic has two hidden fully-connected layers with a size of 512, and an output layer. We train five independent critic networks and use the smallest of the values predicted by critics to train actors, as done in Eysenbach et al. [7].

---

[2]https://github.com/anuragajay/decision-diffuser/tree/main/code
[3]The author thankfully provided a private github repository
[4]https://github.com/ikostrikov/implicit_q_learning
[5]https://github.com/google-research/google-research/tree/master/contrastive_rl
[6]https://github.com/ErickRosete/tacorl

**Diffusion model** The diffusion model for planning milestones is implemented based on the code provided by Ajay et al. [1], which consists of a temporal U-Net architecture with six residual 1-D convolutional blocks. To let the diffusion model predict milestones $g_{1:K}$ based on an initial state $s_0$ and goal state $s_{\text{goal}}$, we fixed $g_0 = f_\omega(s_0)$ and $g_{K+1} = f_\omega(s_{\text{goal}})$ during the denoising process, as done in Janner et al. [21].

**Image preprocessing** Our image preprocessing method used for CALVIN experiment consists of three steps: 1) Resize RGB images size of $200 \times 200 \times 3$ into $128 \times 128 \times 3$. 2) Perform stochastic image shifts of 0-6 pixels. 3) Perform color jitter transform augmentation with a contrast of 0.1, a brightness of 0.1 and hue of 0.02. 4) Normalize value of each pixel to let it falls between $-1$ and 1.

**Visual perception** For CALVIN experiments, we utilize a visual perception network consists of three convolutional layers and a spatial softmax layer. It is the same as the perception network used in Lynch et al. [28] and Rosete-Beas et al. [35].

**Skill encoder and decoder** The skill encoder and decoder are trained using the code provided by Rosete-Beas et al. [35], and have the same architecture as LMP and TACO-RL.

## E   Training details

**D4RL experiments** We train the encoder, goal-conditioned actor, critic and diffusion model simultaneously using a unified loss function: $J_{\text{unified}} := J_{\text{actor}} + J_{\text{critic}} + \alpha J_{\text{diffusion}}$, where the coefficient $\alpha$ is fixed to 0.001 for all the experiments. We train the models for 2.5M training steps for Antmaze environments and 1.0M steps for Kitchen environments. We utilize NVIDIA Geforce RTX 3060 Ti graphics card for training our models, taking approximately 14 hours per 1.0M training steps. We use Adam optimizer [23], with a learning rate of 0.0001.

**CALVIN experiments** We train our model for 1.0M training steps using the same graphics card used for D4RL experiments, taking approximately 40 hours per 1.0M training steps. We use the same optimizer and learning rate used in D4RL experiments.

## F   Hyperparameter setting

In this section, we describe hyperparameter details.

- We set dimension of goal space $\mathcal{G}$ to 16 for Antmaze environments, and 32 for Kitchen and CALVIN environments.
- We use the number of milestones $K = 30$ with maximum interval $\triangle_{\text{max}} = 32$ for Antmaze environments, and $K = 14$ with maximum interval $\triangle_{\text{max}} = 16$ for Kitchen and CALVIN environments.
- We set $\lambda$ in Equation (9) in the main paper to be adaptive to scale of estimated log-likelihood $\log p(g|s, a)$, by setting it as $\lambda = \frac{\tilde{\lambda}}{\frac{1}{|\mathcal{B}|} \sum_{(s,a,g) \in \mathcal{B}} |\log p(g|s,a)|}$ as done in Fujimoto and Gu [9], where $\mathcal{B}$ denotes a mini-batch. We use $\tilde{\lambda}$ of 2.5 for Antmaze environments, and 0.05 for Kitchen and CALVIN environments.
- We use diffusion timestep $N$ of 300, diffusion guidance coefficient $\beta$ of 0.5, and target temporal distance $\triangle_{\text{target}}$ of $0.5\triangle_{\text{max}}$ for the all experiments.
- We use threshold $\delta$ of 0.1 to determine whether the agent has reached the targeted milestone.

## G   Visualization of the learned latent goal space

In this section, we discuss about the ability of DTAMP to learn a latent goal space that captures the dynamics of an environment. We train the encoder along with the actor and critic, through goal-conditioned imitation learning. By doing so, we make the encoder capture useful information from states, to let the actor and critic distinguish the states and actions that can lead an agent to the goal. There are also similar methods that have been studied through prior work, such as learning forward or inverse dynamics model in a latent space [40, 12] or using contrastive learning based on state occupancy measures [7]. Figure 6 visualizes an example of latent goal space learned by DTAMP

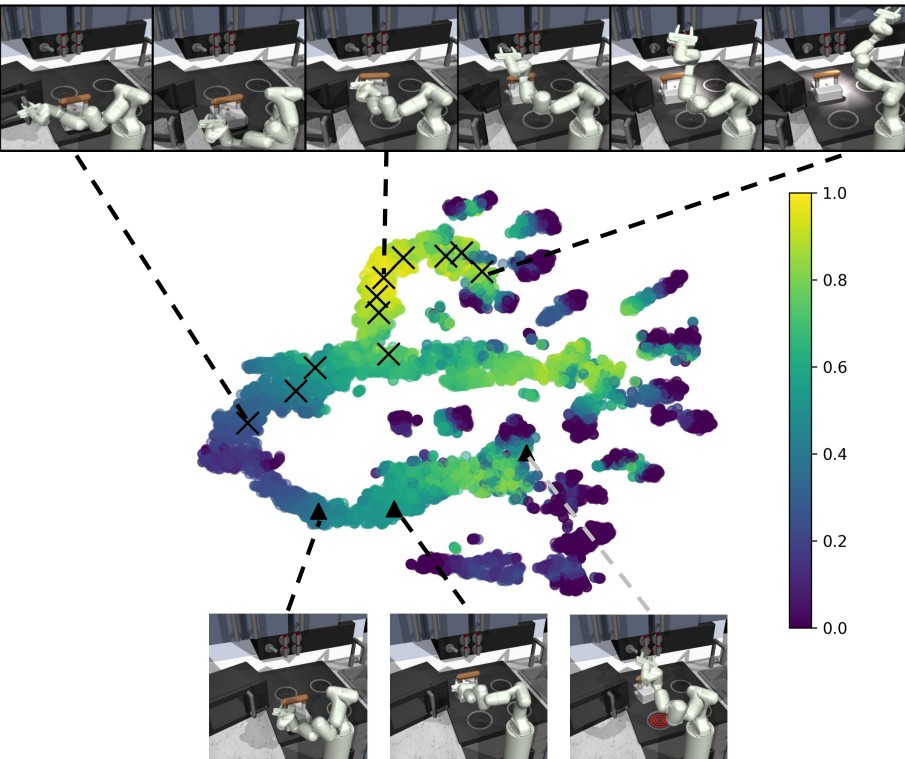

Figure 6: The figure shows t-SNE [43] embedding of the learned latent space. Each data point represents a state in `kitchen-mixed-v0` data. The black x's indicate milestones planned by DTAMP, and the black triangles indicate three states sampled from an arbitrary trajectory. Each image was created by visualizing the data point closest to the corresponding milestone. The colors represent state-values estimated using IQL.

using `kitchen-mixed-v0` data. The figure shows that the trained encoder can capture the dynamics of environment, and also let the milestone planner successfully predict milestones in the learned latent space.

## H    Additional experiments

### H.1    D4RL locomotion tasks

In this experiment, we aim to verify that DTAMP can also deal with the general reinforcement learning problems in which the objectives of the tasks are specified by the reward functions. To this end, we evaluate DTAMP on D4RL locomotion tasks (Halfcheetah, Hopper, and Walker2d). In order to let the milestone planner to predict a trajectory that maximizes the sum of rewards, we made the classifier-free guidance method to be conditioned on sum of rewards as done in Ajay et al. [1]. The comparative result shown in Table 6 shows that DTAMP achieves a marginally higher average score compared to the baselines, which indicates that the proposed method also can be used for general reinforcement learning problems. We would like to note that the D4RL locomotion tasks provide dense rewards and can be performed by planning relatively short trajectories compared to antmaze tasks (to predict more than 100 timesteps forward does not affect much on the performance on the locomotion tasks). This explains why DTAMP does not show a more significant performance improvement over Diffuser and DD in these tasks.

### H.2    Robustness against environment stochasticity

In order to demonstrate the robustness against environment stocahsticity, we further evaluate DTAMP on `maze2d-umaze-v1` environment of the D4RL benchmark, while adding stochasticity to the

| Environment | CQL | IQL | DT | TT | Diffuser | DD | **DTAMP** |
|---|---|---|---|---|---|---|---|
| halfcheetah-medium | 42.6 | 47.4 | 42.6 | 46.9 | 44.2 | **49.1** | 47.3 |
| halfcheetah-medium-replay | 36.6 | 44.2 | 36.6 | 41.9 | 42.2 | 39.3 | **42.6** |
| halfcheetah-medium-expert | 91.6 | 86.7 | 86.8 | **95.0** | 79.8 | 90.6 | 88.2 |
| hopper-medium | 52.9 | 66.3 | 67.6 | 61.1 | 58.5 | 79.3 | **80.7** |
| hopper-medium-replay | 18.1 | 94.7 | 82.7 | 91.5 | 96.8 | **100.0** | **100.0** |
| hopper-medium-expert | 52.5 | 91.5 | 107.6 | 110.0 | 107.2 | **111.8** | 109.4 |
| walker2d-medium | 75.3 | 78.3 | 74.0 | 79.0 | 79.7 | 82.5 | **82.7** |
| walker2d-medium-replay | 26.0 | 73.9 | 66.6 | **82.6** | 61.2 | 75.0 | 79.5 |
| walker2d-medium-expert | 107.5 | 109.6 | 108.1 | 101.9 | 108.4 | **108.8** | 108.2 |
| Average | 77.6 | 77.0 | 74.7 | 78.9 | 75.3 | 81.8 | **82.1** |

Table 6: The table compares the performance of DTAMP evaluated on D4RL locomotion tasks with baselines. The result shows that DTAMP achieves the highest average score, which indicates that the proposed method is also able to handle general reinforcement learning problems as well.

| $p$ | Diffuser | DD | **DTAMP** | Ref. max score |
|---|---|---|---|---|
| 0.0 | 61.9 | 92.6 | **97.6** | 80.5 |
| 0.3 | 29.6 | 97.0 | **99.0** | 72.3 |
| 0.5 | 18.5 | 70.5 | **99.4** | 64.3 |

Table 7: Normalized average score with different levels of stochasticity.

transition model. We also compare DTAMP against Diffuser [21], which predicts future states and actions using a diffusion model, and Decision Diffuser (DD) [1], which utilizes an inverse dynamics model to predict actions from the states planned by a diffusion model. In this experiment we let Diffuser and DD predict the whole trajectory only once at the beginning of each rollout as done in DTAMP.

**Setting**   We create three different levels of stochasticity by choice of $p \in \{0, 0.3, 0.5\}$. The environment either executes a random action $a \sim \text{Unif}(\mathcal{A})$ with probability $p$ or executes the action given by the agent with probability $(1 - p)$. Three sets of offline data were collected separately according to the choice of $p$, and each set of data consists of one million transitions collected by a waypoint controller. We also note that the control frequency is reduced threefold by repeating the same action three times in this experiment. By doing so, we reduce the time horizon of the environment (300 timesteps to 100 timesteps) to let Diffuser and DD predict the whole trajectory only once at the beginning of each rollout.

**Result**   The normalized score of each model is shown in Table 7. The column named Ref. max score (reference max score) shows the average score of the waypoint controller, which represents the performance of an optimal agent. We note that even if an agent predicts optimal actions, stochasticity of environment causes inherent performance degradation. In order to neglect this effect and only compare the robustness of each model, the scores shown in Table 7 are normalized respect to the reference max scores (*i.e.*, normalized score $= 100 \times \frac{\text{average score}}{\text{ref. max score}}$).

The result shows that DTAMP is robust against the environment stochasticity and does not show performance degradation in the terms of normalized score. On the other hand, Diffuser shows greater performance degradation as stochasticity increased. DD shows no performance degradation when the stochasticity is small ($p = 0.3$) while showing performance degradation when the stochasiticity is large ($p = 0.5$). The result indicates that our approach using a goal-conditioned actor makes DTAMP robust against environment stochasticity and allows to plan milestones only once for each rollout, while the other diffusion based planners do not.

