# OpenReview forum: "Diffused Task-Agnostic Milestone Planner"
_NeurIPS.cc/2023/Conference — NeurIPS 2023 poster_

### Official Review · Reviewer_xaPC · 2023-07-02

**Soundness:** 4 excellent
**Presentation:** 4 excellent
**Contribution:** 4 excellent
**Rating:** 8
**Confidence:** 3

**Summary:**

This paper proposed a novel method to solve long-horizon, sparse-reward tasks and multi-task problems, which outperforms offline RL methods on many benchmarks.

**Strengths:**

- Provide an elegant and general idea to solve sparse-reward problem -- which is hard for RL-based methods.
- The paper is well-written and easy to be understood.
- Both theoretical proof and empirical details are provided, which make the claims in the paper persuasive.
- Good results.

**Weaknesses:**

- How much do the sampling methods matter? In the paper, the number of sampled points are fixed. If the authors can provide more ablation results with varying number of sampling points or use other sampling methods, the results will be more instructive.

**Questions:**

Mentioned in weakness.

**Limitations:**

No obvious societal impact.

---

> ### Author Rebuttal · Authors · 2023-08-09
>
> We thank the reviewer for the valuable comments including the positive comment on the soundness of our paper and the suggestion of ablation study to make our manuscript more instructive.
> We present the responses to the reviewer's concerns and questions below.
>
> **Q. If the authors can provide more ablation results with varying number of sampling points or use other sampling methods, the results will be more instructive.**
>
> Thank you for your suggestion of ablation study on the number of sampled milestones to make the results more instructive.
> To reflect the reviewer's comments, we constructed additional experiments to determine how changes in the number of milestones affect performance.
> The results of the additional experiments are presented in the attached PDF file.
> The results show that using the small number of milestones leads to marginal degradation of performance.
> This is because the temporal distance between milestones increases as the number of milestones decreases, which results to increase the burden of the goal-conditioned actor to reach distant subgoals.
> On the other hand, if the number of milestones exceeds a certain threshold (30 milestones for antmaze-medium task), the change in the number does not affect performance of DTAMP.
>
> **Q. No obvious societal impact.**
>
> A. The proposed method can be applied to a wide range of robotic tasks that require long-term planning in our society, e.g. autonomous driving, indoor navigation.
> In addition, the advantages of our method of performing multiple tasks with a single agent can lead to cost savings in industrial robot development.

---

> > ### Comment · Reviewer_xaPC · 2023-08-14
> >
> > Thanks for the detailed explanation!

---

> > > ### Author Response · Authors · 2023-08-21
> > >
> > > Thank you for acknowledging our contribution on proposing a method to solve long-horizon, sparse-reward problems.
> > > Your discussion was also helpful for conducting additional ablation studies which make our paper more instructive.

---

### Official Review · Reviewer_M7TA · 2023-07-07

**Soundness:** 2 fair
**Presentation:** 3 good
**Contribution:** 3 good
**Rating:** 6
**Confidence:** 3

**Summary:**

This paper extends diffusion-based latent milestone planners to long-term planning, vision control, and multi-task settings. Specifically, an encoder is trained to project observations into the latent space. The authors employ goal-conditioned imitation learning to train both the encoder and the prior goal-conditioned policy. After that, a diffusion probabilistic model is trained to model the milestone trajectories.

In order to generate the shortest path leading to the goal state more efficiently, the authors take into account the temporal distance between consecutive milestones in addition to the initial state and goal state. At the low level, the prior goal-conditioned policy is used to sample atomic actions, guiding the agent toward the given milestone.

Experiments conducted on selected tasks from the Deep RL for Real Life (D4RL) and CALVIN benchmarks demonstrate superior performance over the baseline models.

**Strengths:**

This paper is well-organized. They proposed shortest-path guidance method is effective.

**Weaknesses:**


- I think the current experiments are not enough. Could the authors also provide results on the D4RL MuJoCo locomotion tasks? It would strengthen the paper if better performance could also be achieved there.
- I'm wondering why the PointMaze tasks were not selected for comparison with the results from the Diffuser[1].
- Can the authors ablate on the choice of image encoder in the image-based planning to verify that joint train a goal-conditioned actor and critic is necessary?



[1]: Michael Janner, Yilun Du, Joshua Tenenbaum, and Sergey Levine. Planning with diffusion for flexible behavior synthesis. In Proceedings of the International Conference on Machine Learning, Baltimore, US, Jul 2022.

**Questions:**

- Why are there only three different goals in the multi-goal setting? Isn't the initial state and goal state assigned manually and the goal-conditioned plans are sampled with inpainting?

**Limitations:**

Yes.

---

> ### Author Rebuttal · Authors · 2023-08-09
>
> We thank the reviewer for the valuable comments including acknowledging our contribution on proposing an effective method to generate shortest path and suggestions to strengthen our paper.
> We present the responses to the reviewer's concerns and questions below.
>
> **Q. Could the authors also provide results on the D4RL MuJoCo locomotion tasks?**
>
> A. Thank you for your suggestion to improve the impact of our work.
> To reflect the reviewer's comment, we conducted further experiments on the mentioned tasks by modifying the diffusion guidance method to maximize the sum of rewards rather than to minimize temporal distance between milestones.
> The attached PDF file presents the performance of DTAMP evaluated in the D4RL locomotion tasks, which achieves a marginally higher average score compared to the baselines.
> We would like to note that the main purpose of our approach is to address long-horizon, sparse-reward problems and image-based tasks which the existing diffusion-based sequence modeling methods (Diffuser and Decision Diffuser) cannot handle.
> However, the D4RL locomotion tasks provide dense rewards and can be performed by planning relatively short trajectories compared to antmaze tasks (to predict more than 100 timesteps forward does not affect much on the performance on the locomotion tasks).
> This explains why DTAMP does not show a more significant performance improvement over Decision Diffuser in these tasks.
> Meanwhile, we would like to emphasize that the greater contribution of our paper is to broaden the field where generative flexibility of diffusion models can be exploited, rather than performing better in the problems already covered by the existing diffusion-based models.
>
> **Q. I'm wondering why the PointMaze tasks were not selected for comparison with the results from the Diffuser.**
>
> A. We have conducted experiment on a Pointmaze environment (U-maze env.) and presented the result in our supplementary material (see Section H of the supplementary material).
> In this experiment, we added three different levels of stochasticity to the system dynamics to demonstrate DTAMP's robustness against environment stochasticity.
> As a result, DTAMP achieved the score close to the maxium possible value at all three levels of stochasticity.
> The reason why we did not further evaluated the proposed method on the other variations of Pointmaze environment (medium and large maze) is that we believe that the experimental results on Antmaze tasks (in Table 2 and Table 3 of our paper) are sufficient to show that DTAMP can plan trajectories for various goal positions in maze of various sizes.
> This is because the Pointmaze tasks can be solved much more easily than the Antmaze tasks as the Pointmaze tasks take less timesteps to reach their goals and have a smaller action space, making point robots easier to control than ant robots.
> We would like to further mention that the papers of Contrastive RL [1], Decision Diffuser [2], and Trajectory Transformer [3] also do not provide the performance of their algorithms on the mentioned environments.
> This fact makes it challenging to fairly compare DTAMP with baselines on Pointmaze environments, as the environment specific setting of hyper-parameters might be needed to accurately evaluate the baseline methods.
>
> [1] Eysenbach et. al., "Contrastive Learning as a Goal-Conditioned Reinforcement Learning", NeurIPS 2022
> [2] Ajay et. al., "Is Conditional Generative Modeling All You Need for Decision-Making?", ICLR 2023
> [3] Janner et. al., "Offline Reinforcement Learning as One Big Sequence Modeling Problem", NeurIPS 2021
>
> **Q. Can the authors ablate on the choice of image encoder in the image-based planning?**
>
> A. Thank you for your suggestion to clarify our contribution on proposing a method to learn latent milestone representation.
> To reflect the reviewer's comment, we conducted an additional experiment of DTAMP using a variational autoencoder (VAE) as an image encoder instead of training the encoder jointly with actor and critic.
> The result of the ablation study is presented in the attached PDF file.
> Our empirical analysis shows that the representations learned by VAE cannot capture dynamical distances (distance in terms of how far apart are states in timesteps),
> which results in prediction of infeasible trajectories by DTAMP.
>
> **Q. Why are there only three different goals in the multi-goal setting?**
>
> A. We found that increasing the number of different goals increases the variance of success rates and requires more rollouts for accurate evaluation.
> Especially, computational burden of Trajectory Transformer makes it difficult to simulate a sufficient number of rollouts (about an hour per episode).
> Therefore, we selected three goals that can include as diverse paths as possible on the maze.

---

> > ### Comment · Reviewer_M7TA · 2023-08-17
> > **Response**
> >
> > Thank the authors for the additional experiments! I'm satisfied with the results, and I'm happy to increase my score.

---

> > > ### Author Response · Authors · 2023-08-21
> > >
> > > Thank you for acknowledging that the proposed method can also handle general reinforcement learning problems. We also thank you for the valuable comments which were helpful for conducting additional ablation studies.

---

> ### Comment · Area_Chair_CYzr · 2023-08-13
> **Additional Experiments**
>
> Dear Reviewer,
>
> Please look at the authors' rebuttal, especially with regard to additional experiments, and see whether that would change your rating.
>
> Thanks,
>
> Your AC

---

### Official Review · Reviewer_8zhN · 2023-07-08

**Soundness:** 3 good
**Presentation:** 2 fair
**Contribution:** 3 good
**Rating:** 6
**Confidence:** 4

**Summary:**

This paper introduces a hierarchical architecture named DTAMP to solve sequential decision-making problems. Specifically, the high-level part of the architecture is realized by a diffusion model to decompose the long-term goal into several short-term milestones. Then, the low-level part makes basic decisions at fixed intervals until the milestone is reached, and the policy is trained by goal-conditioned imitation learning. The results show that the proposed approach can achieve state-of-the-art performance on D4RL and CALVIN benchmarks.

**Strengths:**

1. The method combines the diffusion model with hierarchical reinforcement learning, which can achieve state-of-the-art performance on multiple benchmarks.
2. The intermediate latent goals, i.e., milestones obtained by the proposed approach, can be represented not only by traditional discrete labels but also by images, as shown in Figure 4.

**Weaknesses:**

1. The biggest concern would be that the training process is not clearly explained. The authors do not state whether they trained the diffusion model and imitation learning separately or synchronously. If they're trained separately, how to obtain the labeled data of milestones? The authors should supplement enough information.
2. In Table I, the target interval value denotes the target temporal distance between milestones, which is set to be relatively smaller than the maximum value. Then, there is no restriction on the target interval value when the ratio is 1.0, which should be the same as in the unconditioned experiment. The authors should make a further clarification on the increased timesteps when the ratio is 1.0.
3. The structural relationship of Figure 1. is quite confusing. According to my understanding, the diffuser (as shown in the right subfigure) should be performed first, and then the actor and critic (as shown in the left subfigure) should be performed to obtain the trajectory. It is recommended to arrange the subfigures from left to right and establish the relationship between the two sub-figures.
4. There are some minor mistakes in this paper, such as typos in Line 122, 176, and 'inference time' in Figure 3.


**Questions:**

1. The authors state that the trained actor is limited in inferring a suitable action for a distant goal if the offline data does not contain a trajectory connecting the current state and the goal state at the end of Section 3.1. Then, when the amount of data decreases, will the performance of the proposed approach decrease significantly?
2. In this paper, the number of planned intervals K is a fixed value. Will different values of K have a significant impact on performance, and is there a situation where the milestone cannot be reached after K intervals?


**Limitations:**

See above.

---

> ### Author Rebuttal · Authors · 2023-08-09
>
> We thank the reviewer for valuable comments including summary of our contributions and pointing out some of our explanations that was not clear enough.
> We present the responses to the reviewer's concerns and questions below.
>
> **Q. The biggest concern would be that the training process is not clearly explained.**
>
> A. We trained the diffusion model and imitation learning synchronously by adding up the losses as,
> $J_\text{unified}=J_\text{actor} + J_\text{critic} + \alpha J_{diffusion}$,
> where the coefficient $\alpha$ is fixed to 0.001 for all the experiments.
> We would like to mention that it is presented in Section E (Training details) of our supplementary material.
>
> **Q. The authors should make a further clarification on the increased timesteps when the ratio is 1.0.**
>
> A. We would like to note that setting a ratio of 1.0 leads to planning the longest path by making the temporal distance between milestones the maximum value $\triangle_\text{max}$.
> On the other hand, in the case of planning without condition of target temporal distance, the generated path can have various lengths from the shortest length to the longest length.
> As a result, the trajectories planned under the condition of $\triangle_\text{target}/\triangle_\text{max}=1$ shows longer average path-length than the trajectories planned unconditionally.
> We will clarify how the setting of target temporal distance affects the length of generated trajectory in the revised version of our manuscript, to avoid the possible confusion.
>
> **Q. The structural relationship of Figure 1. is quite confusing.**
>
> A. We wanted to illustrate the training process of DTAMP in Figure 1.
> We thought that we should explain how to learn milestone representations first, and then explain how to train the diffusion model to plan the learned milestones.
> However, we understand the confusion and we will rearrange and revise the figure for the final version of our paper to more clearly describe how the proposed method works.
>
> **Q. There are some minor mistakes in this paper, such as typos in Line 122, 176, and 'inference time' in Figure 3.**
>
> A. We appreciate for pointing out the typos we missed.
> It will help a lot to revise our manuscript.
>
> **Q.  When the amount of data decreases, will the performance of the proposed approach decrease significantly?**
>
> A. No, the statement at the end of Section 3.1 does not mean that the performance of DTAMP will decrease significantly when the amount of data decreases.
> It means that even if the trained actor may suffer to predict a suitable action for a distant goal, the proposed diffusion-based milestone planner provides milestones that can guide the actor, so that enables the actor to eventually reach the distant goal by following the relatively close milestones.
> We understand the confusion and will clarify how the proposed milestone planner enhances sample efficiency for the long-horizon, sparse-reward tasks in the revised version of our manuscript.
>
> **Q. Will different values of K have a significant impact on performance, and is there a situation where the milestone cannot be reached after K intervals?**
>
> A. We agree that there is a lack of explanation for how changes in the number of milestones affect performance.
> To answer to the reviewer's question, we constructed additional experiments to determine how changes in the number of milestones affect performance. The results of the additional experiments are presented in the attached PDF file. The results show that using the small number of milestones leads to marginal degradation of performance. This is because the temporal distance between milestones increases as the number of milestones decreases, which results to increase the burden of the goal-conditioned actor to reach distant subgoals. On the other hand, if the number of milestones exceeds a certain threshold (30 milestones for antmaze-medium task), the change in the number does not affect performance of DTAMP.
> From this empirical analysis, we can conclude the issue mentioned by the reviewer can be resolved by setting the sufficient number of milestones that covers the time horizon of the targeted environment.

---

### Official Review · Reviewer_d8Sq · 2023-07-10

**Soundness:** 3 good
**Presentation:** 3 good
**Contribution:** 3 good
**Rating:** 5
**Confidence:** 1

**Summary:**

This paper uses sequence modeling method in applications like long-term planning, vision-based control, and multi-task decision-making. They formulate a novel method which uses diffusion-based generative sequence model to plan a series of milestones in a latent space and to have an agent to follow the milestones to get the task done. Their method can learn control-relevant, low-dimensional latent representations of milestones that makes it possible to efficiently perform long-term planning and vision-based control.

They train an encoder to extract milestone representations from observations, and an actor to predict actions to follow the milestones, using a goal-conditioned imitation learning fashion. Their method uses a denosing diffusion model to generate an array of milestones conditioned on a target state. The minimum temporal distance diffusion guidance method is used in this paper to make the proposed method plan the shortest path to reach the target state.

**Strengths:**

1) The encoder trained using the method proposed, encodes control-relevant features into unique latent representations, which lets efficient planning even if the observations are high-dimensional.
2) The diffusion-based planner enables flexible planning for a variety of tasks and takes advantage on multi-task decision-making problems.
3) Using the trained actor as a feedback controller makes an agent robust against the environment stochasticity, and makes it possible to plan milestones only once at the beginning of an episode which largely reduces inference time compared to the existing sequence modelling methods. DTAMP has lower inference time compared to other baselines.
4) DTAMP outperforms the baseline methods on the every environment.

**Weaknesses:**

1) Language of the paper could be improved. Some of the words are repetitive and there are quite some typos. for eg: "letting the the diffusion " and many other - at other places.
2) DTAMP: the ablation study on how the design of diffusion model was done is missing.



**Questions:**

1) Did the authors re-implement all the baselines mentioned in Table2 for the comparison?

2) why and how did authors take the decisions to built the diffusion model how they did?


**Limitations:**

 1) The proposed method might not work on the goal-reaching tasks and authors have not used it for the task not having goal reaching objective.
2) Their method cannot solve a new task not included in dataset.

Authors can also elaborate on societal impact of their work.

---

> ### Author Rebuttal · Authors · 2023-08-09
>
> We thank the reviewer for the valuable comments including summary of our contributions and pointing out typos we missed.
> We present the responses to the reviewer's concerns and questions below.
>
> **Q. Language of the paper could be improved.**
>
> A. Thank you for your valuable comment for improving the quality of our paper, and we will refer to it for revision.
>
> **Q. Did the authors re-implement all the baselines mentioned in Table2 for the comparison?**
>
> A. We did not re-implement the baseline algorithms.
> The performances of baselines are mostly taken from their original paper, except we evaluated Diffuser, Decision Diffuser and Trajectory Transformer on Antmaze environments using the code provided by the authors of their papers.
> For a more detailed description of the source of baseline performance, please refer to Section C (Source of baseline performance) of the supplementary material.
>
> **Q. Why and how did authors take the decisions to built the diffusion model how they did?**
>
> A. We used a network architecture of the same diffusion model as Diffuser and Decision Diffuser for fair comparison.
> We made this decision to verify that our approach can broaden the area of application of diffusion-based sequence modeling methods without changing the architecture of the diffusion model.
> For a more detailed description of implementation details, please refer to Section D (Implementation details) of the supplementary material.
>
> **Q. The proposed method might not work on the task not having goal reaching objective.**
>
> A. We conducted further experiments on the D4RL locomotion tasks (Halfcheetah, Hopper, and Walker2d) which provide dense rewards and should be handled by maximizing "sum-of-trajectory-rewards" objective. The additional experiments were done by modifying the diffusion guidance method to maximize the sum of rewards rather than to minimize temporal distance between milestones. The attached PDF file presents the performance of DTAMP evaluated in the D4RL locomotion tasks, which achieves a marginally higher average score compared to the baselines. This result indicates that DTAMP can also handle the general reinforcement learning problems as well with a small modification.
>
> **Q. Authors can also elaborate on societal impact of their work.**
>
> A. The proposed method can be applied to a wide range of robotic tasks that require long-term planning in our society, e.g. autonomous driving, indoor navigation.
> In addition, the advantages of our method of performing multiple tasks with a single agent can lead to cost savings in industrial robot development.

---

> > ### Comment · Reviewer_d8Sq · 2023-08-17
> >
> > Thank you for the answers. I am happy to keep my current rating.

---

> > > ### Author Response · Authors · 2023-08-21
> > >
> > > Thank you for acknowledging our contribution on developing diffusion-based planning that enables flexible planning and takes advantage on multi-task decision-making problems. Also, your comments were really helpful for revising our paper.

---

### Official Review · Reviewer_g1BQ · 2023-07-10

**Soundness:** 3 good
**Presentation:** 3 good
**Contribution:** 3 good
**Rating:** 5
**Confidence:** 4

**Summary:**

The paper studies the problem of latent-space planning for sequential decision-making tasks using generative diffusion models. Similarly to works such as Diffuser, they train an endpoint-conditionned generative model to generate sub-goals along a path starting at a given state and reaching a certain goal. Differently to Diffuser, they perform the generation in a latent space pre-trained with imitation learning. The method achieves good results on D4RL environments as well as on a pixel-based planning task.

**Strengths:**

* The paper is well-written and, for someone who is not expert in planning or diffusion models, can be very easy to understand.
* The proposed approach makes sense in the current context, i.e. planning to visit subgoals in long-horizon tasks. I appreciate the approach and specifically the contribution related to this line of work, as it reduces advances in the field of planning and/or control to advances in the field of generative models (i.e. better diffusion models -> better goal trajectories -> better performance to some extent).
* Experimental results show that the proposed method has competitive performance with baselines and latest SOTA, both on state-based and pixel-based domains.

**Weaknesses:**

* I am not sure how exactly the approach differs from Diffuser (Janner et al, 2022), at least in the second phase when generating subgoals? From what I understand, the difference is in the first phase, where a latent space with desirable features for control is learned.
* Based on Figure 4 of the Decision Diffuser paper, the method achieves high returns on the D4RL locomotion task, while results from your Table 2 paper seem to indicate otherwise. Can you explain this discrepancy in results?
* I am curious how the method performs against recent generative-model-based model-free algorithms such as the  contrastive value estimation line of work (Contrastive Learning As a Reinforcement Learning Algorithm by Eysenbach et al. 2023 and Contrastive Value Learning: Implicit Models for Simple Offline RL by Mazoure et al. 2023)?

**Questions:**

* Can you elaborate on the distinction of your DTAMP method vs the Diffuser algorithm?
* If possible, can you show a comparison against the method from “Contrastive Learning As a Reinforcement Learning Algorithm”?
* Can you elaborate on the discrepancy in results between Decision Diffuser paper and your Table 2?
* How was the noise schedule chosen for various experiments? If you perform state-based denoising, even in a latent space, I would imagine its structure to be somewhat different to the latent space of a pixel-based tasks.

**Limitations:**

Main limitation is that the empirical contribution and evaluation is focused on goal-conditionned tasks, which reduces to maximizing the 0-1 sparse reward function, as opposed to the general “sum-of-trajectory-rewards” objective.

---

> ### Author Rebuttal · Authors · 2023-08-09
>
> We thank the reviewer for the valuable reviews including the positive comment on legibility.
> We present the responses to the reviewer's concerns and questions below.
>
> **Q. Can you elaborate on the distinction of your DTAMP method vs the Diffuser algorithm?**
>
> A. The main distinctive features of DTAMP compared against Diffuser are learning latent representation of subgoals (we named it milestones) and generating temporally sparse milestones instead of predicting the every states and actions connecting the current state and the goal.
> It is an intuitive idea and easy to be built on an implementation of Diffuser, while providing large benefits in terms of reducing computational costs and making it possible to perform long-horizon and partially observable tasks which Diffuser cannot handle.
> Furthermore, proposing a diffusion guidance method to ensure the planned trajectory to be the shortest path to the goal is another contribution of our work, while Diffuser does not guarantee the planned trajectory to be the shortest.
>
> **Q. Can you show a comparison against the method from “Contrastive Learning As a Reinforcement Learning Algorithm”?**
>
> A. We already compared DTAMP against the method from the mentioned paper [1].
> The algorithm is indicated by "ContRL" in Table 2 and Table 3 of our manuscript.
> While ContRL also utilizes a bootstrapping-free method to train a critic function, it shows limited performance in long-horizon tasks such as Antmaze environments.
> On the other hand, our approach to generate a series of milestones divides a long-term problem into short-term problems and let an agent solve them more easily without using the bootstrapping method.
>
> [1] Eysenbach et. al., "Contrastive Learning as a Goal-Conditioned Reinforcement Learning", NeurIPS 2022
>
> **Q. Can you elaborate on the discrepancy in results between Decision Diffuser paper and your Table 2?**
>
> A. The performance of the Decision Diffuser shown in Figure 4 of its paper [2] was evaluated on D4RL locomotion tasks (halfcheetah, hopper, and walker2d), and the results in Table 2 of our paper were evaluated on D4RL antmaze tasks.
> The D4RL locomotion tasks provide dense rewards (depending on the robot's velocity and posture at each time step), while the antmaze tasks provide a reward of +1 only when the robot reaches the goal.
> When dense rewards are provided, an agent can achieve high performance by planning relatively short trajectories (planning trajectories of about 100 timesteps is sufficient to accomplish the locomotion tasks), while antmaze tasks require an agent to plan trajectories of longer than 500 timesteps.
> As a result, Decision Diffuser exhibits poor performance in antmaze tasks due to its limited ability to predict long-term trajectories.
>
> [2] Ajay et. al., "Is Conditional Generative Modeling All You Need for Decision-Making?", ICLR 2023
>
> **Q. How was the noise schedule chosen for various experiments?**
>
> A. We utilized the same noise schedule (cosine schedule) used by Diffuser and Decision Diffuser for fair comparison, and the same scheduling method was used for demonstrating DTAMP on image-based environments.
> It is worth mentioning that the output of the encoder is normalized in our algorithm, so that the signal-to-noise ratio is consistent across various domains, which reduces the burden of choosing a different scheduling method for each different environment.
> We agree that there was a lack of explanation about how we designed the diffusion model, and we will include it in the final version of our paper.
>
> **Q. Main limitation is that the empirical contribution and evaluation is focused on goal-conditioned tasks.**
>
> A. We conducted further experiments on the D4RL locomotion tasks (Halfcheetah, Hopper, and Walker2d) which provide dense rewards and should be handled by maximizing "sum-of-trajectory-rewards" objective. The additional experiments were done by modifying the diffusion guidance method to maximize the sum of rewards rather than to minimize temporal distance between milestones.
> The attached PDF file presents the performance of DTAMP evaluated in the D4RL locomotion tasks, which achieves a marginally higher average score compared to the baselines.
> This result indicates that DTAMP can also handle the general reinforcement learning problems as well with a small modification.

---

### Official Review · Reviewer_Haes · 2023-07-15

**Soundness:** 3 good
**Presentation:** 3 good
**Contribution:** 3 good
**Rating:** 6
**Confidence:** 3

**Summary:**

The paper proposes uses a diffusion-based generative model to plan a sequence of milestones in a latent space and have the agent follow this latent plan to accomplish a given task. The authors show  results on AntMaze, and more importantly on the CALVIN Benchmark to show the effectiveness of their method for image-based planning.

**Strengths:**

- The direction of using diffusion-model based generative models for planning is definitely quite interesting. I also agree with the authors that generating the plan in a learned latent space is a promising direction.
- I also thought that the use of classifier-free diffusion guidance to encourage the trajectory follow the shortest path was pretty neat.
- I appreciate the authors presenting experiments on a partially observed environment like CALVIN! Showing that their method can come up with a reasonable latent plan for partially observed environment is promising.
- I also liked that the authors tried visualising the learned latent space. (Fig 2. in Supp). The visualisation shows how the proposed approach, can plan successfully in the learned latent space to sequentially accomplish multiple tasks in the environment.

**Weaknesses:**

- In L175, the authors mention that their approaches can only do the diffusion step at the beginning once, compared to existing methods which have to predict future trajectories at every step. Can the authors describe how can the policy recover from failure to reach a `milestone' in their case?  If the agent's current state and the desired state is never closer than the given threshold, then according to the Alg. 1, the agent will never recover from this.
- Can the authors comment on the performance of their approach will work when both the environment, and the state space is more complex and partially observed? Something that comes to mind is ImageNav in indoor environments [1] [2]. These environment seem more challenging since trajectories are often longer, and predicting intermediate milestones is much harder than CALVIN like benchmarks.
- For Robustness against environment stochasticity experiments in the supplementary, isn’t the comparison to DD, and Diffuser a bit unfair? The experiments compare to DD and Diffusion by only allowing them to only predict the whole trajectory at once. But inference time related issues aside, DD and Diffuser model which predict the future trajectory at each timestep should be more robust to stochasticity in the environment than the methods which predict the whole trajectory at the beginning.

**Questions:**

- For an exhaustive one-to-one comparisons with existing methods, it’d be good to show experiments for HalfCheetah, Hopper, and Walker 2D as done in both Janner et al, and Ajay et al. Additionally, these aforementioned papers also include block stacking experiments which aren’t included in this manuscript.

- I only partially understood Fig 2 in Supplementary. How did the authors generate corresponding images for certain milestones (X) in the planned trajectory? Are these images corresponding to the state the agent reached when it was "closest" to the milestone?

---

> ### Author Rebuttal · Authors · 2023-08-08
>
> We thank the reviewer for valuable comments, including acknowledging the contribution of our work and pointing out unclear explanations.
> We present the responses to the reviewer's concerns and questions below.
>
> **Q. Can the authors describe how can the policy recover from failure to reach a `milestone' in their case?**
>
> A. To avoid the mentioned issue, we set a time limit and let an agent to move on to the next milestone when it fails to reach the current one within the time limit.
> We found that the performance of DTAMP is not significantly affected by the small changes in the setting of time-limit and setting the time limit of about twice the maximum interval $\triangle_{max}$ was sufficient for our experiments.
> We agree that there was a lack of explanation, and we will include it in the final version of our paper.
>
> **Q. Can the authors comment on the performance of their approach will work when both the environment, and the state space is more complex and partially observed? (e.g. ImageNav)?**
>
> A. Experimental results in antmaze environments show that DTAMP can handle long-horizon tasks that require an agent to plan paths of up to a thousand of timesteps.
> Therefore, DTAMP would be able to successfully plan the trajectories for the environments that have even longer time horizon than CALVIN environment.
> In our opinion, learning the milestone representation of images obtained from changing camera poses will be a major challenge for applying DTAMP to visual navigation domain such as ImageNav environment.
> Meanwhile, there are existing studies proposing promising ways to learn latent representation that capture 3D scenes by encoding radiance fields and utilize it for indoor navigation [1][2].
> Combined with these methods, DTAMP will also be able to address the challenging visual navigation problems as well.
>
> [1] Bautista et. al., "GAUDI: A Neural Architect for Immersive 3D Scene Generation", NeurIPS 2022
> [2] Kwon et. al., "Renderable Neural Radience Map for Visual Navigation", CVPR 2023
>
> **Q. For Robustness against environment stochasticity experiments in the supplementary, isn’t the comparison to DD, and Diffuser a bit unfair?**
>
> A. We would like to note that the experiment of Table 1 in our supplementary material was performed to explain why DTAMP allows an agent to predict the trajectory only once whereas Diffuser and DD cannot.
> To that end, we think the performed experiment sufficiently supports our claim:
> the learned goal-conditioned actor can adapt to stochastic transitions and allows us to perform the time-consuming denoising process only once at the beginning of an episode (in Section 3.4 L174).
> In addition, the result in Table 2 in our supplementary material demonstrates that allowing DTAMP to replan milestones during an episode makes it more robust to stochasticity and enhances its performance.
> However, we understand the confusion and will clarify the purpose of the experiment the revised version of our manuscript.
>
> **Q. For an exhaustive one-to-one comparisons with existing methods, it’d be good to show experiments for HalfCheetah, Hopper, and Walker2D.**
>
> A. Thank you for your suggestion to improve the impact of our work.
> To reflect the reviewer's comment, we conducted further experiments on the mentioned tasks by modifying the diffusion guidance method to maximize the sum of rewards rather than to minimize temporal distance between milestones.
> The attached PDF file presents the performance of DTAMP evaluated in the D4RL locomotion tasks, which achieves a marginally higher average score compared to the baselines.
> We would like to note that the main purpose of our approach is to address long-horizon, sparse-reward problems and image-based tasks which the existing diffusion-based sequence modeling methods (Diffuser and Decision Diffuser) cannot handle.
> However, the D4RL locomotion tasks provide dense rewards and can be performed by planning relatively short trajectories compared to antmaze tasks (to predict more than 100 timesteps forward does not affect much on the performance on the locomotion tasks).
> This explains why DTAMP does not show a more significant performance improvement over Decision Diffuser in these tasks.
> Meanwhile, we would like to emphasize that the greater contribution of our paper is to broaden the field where generative flexibility of diffusion models can be exploited, rather than performing better in the problems already covered by the existing diffusion-based models.
>
> **Q. How did the authors generate corresponding images for certain milestones (X) in the planned trajectory?**
>
> A. To visualize each milestone, we selected a state in the "kitchen-mixed-v0" dataset, which is the closest to each milestone in the latent space.
> We thank the reviewer for pointing out unclear explanation, and we will refer to it for revising the paper.

---

> > ### Comment · Reviewer_Haes · 2023-08-15
> > **Thank you for your responses!**
> >
> > Thank you for responding to my concerns. I am satisfied by the answers and I am happy to keep my rating.

---

> > > ### Author Response · Authors · 2023-08-21
> > >
> > > Thank you for acknowledging our contribution on proposing a diffusion model based generative model which plans trajectory in a latent space. We also thank you for your valuable discussion which will help us for advancing our work in the future.

---

### Author Rebuttal · Authors · 2023-08-09

Dear all the reviewers.

We thank the reviewers for acknowledging the contributions of our work and for making constructive comments to improve the submitted manuscript.
We are pleased that reviewers find out the major strengths of the proposed method, which can be summarized as follows:
* Planning trajectories in the learned latent space using a diffusion model is interesting and promising way to solve partially observable problems.
* The proposed diffusion guidance method to predict the shortest path to a given goal is effective and neat.
* The proposed framework to use a goal-conditioned actor makes an agent robust against the environment stochasticity and largely reduces inference time.
* The proposed method shows state-of-the-art performance on the long-horizon, sparse-reward tasks and an image-based, multi-task benchmark.
* The paper is well-written and easy-to-understand.

We also find that the majority of the reviewers' comments can be summarized in two folds:
* Additional experiments would make the paper more instructive (demonstration on the locomotion tasks, ablation studies to investigate how the number of milestones affects the performance, etc.).
  - We constructed several additional experiments to reflect the reviewers' suggestions, and present the results in the attached PDF file. We hope that the extended empirical analysis will enrich our paper and resolve the reviewers' concerns.
* There are a few unclear explanations and typos that need to be revised.
  - We are sincerely appreciate for pointing out the shortcomings of our presentation. We will actively reflect the advice for revising the manuscript.

Finally, we thank again the reviewers for their valuable reviews and hope that our responses will be sufficient answers to their concerns and questions.

---

### Decision · Program_Chairs · 2023-09-21

**Decision:**

Accept (poster)

**Comment:**

The paper is concerns with using diffusion-based models to generate multiple future milestones that are subsequently used for planning in control tasks.

The paper has received 1 x Strong Accept, 3 x Weak Accept, 2 x Borderline Accept.

All reviewers are exciting in seeing the application of diffusion models to control tasks and appreciate the exact formulation the authors propose — predicting long horizon trajectories in latent space. They find the approach novel (closest one is Diffuser but the current work does operate in latent space which is appreciated), and effective (the reviewers appreciate the results presented on several benchmarks, both more simple as well as pixel-based ones; also ones with partially observable spaces). The approach is also considered timely and promising research area (diffusion models for planning). Hence, the AC in consultation with SAC accepts the paper to Neurips.

The authors are encouraged to add the clarifications from the rebuttal (a more clear explanation of differences to Diffuser, explanation of the results are asked, clarification of the training procedure, additional D4RL results).